# The Paradox of AI Empowerment in Primary School Physical Education: Why Technology May Hinder, Not Help, Teaching Efficiency

**DOI:** 10.3390/bs15020240

**Published:** 2025-02-19

**Authors:** Haoran Zha, Wenye Li, Weihao Wang, Jian Xiao

**Affiliations:** 1College of Physical Education and Health Management, Chongqing University of Education, Chongqing 400065, China; zhahr@cque.edu.cn; 2Institute of Education, Nanjing University, Nanjing 210093, China; 3The Faculty of Education, East China Normal University, Shanghai 200241, China; 52274110003@stu.ecnu.edu.cn (W.W.); 52284110005@stu.ecnu.edu.cn (J.X.)

**Keywords:** artificial intelligence, physical education, teaching efficiency, socio-technical systems theory, classroom management, technology integration, organizational barriers, teacher perspectives

## Abstract

This study investigates why artificial intelligence (AI) may hinder rather than enhance teaching efficiency in primary school physical education (PE). Guided by socio-technical systems theory, we conducted focus group interviews with 13 PE teachers (6 from Nanjing and 7 from Chongqing, China) who had at least three years of teaching experience and two years of AI implementation experience. Participants were purposefully selected through a two-stage sampling strategy: initial screening via open-ended questionnaires to identify teachers reporting negative experiences with AI integration, followed by snowball sampling to recruit additional participants with similar perspectives. Data collection employed a dual-facilitator approach using semi-structured interviews, with one moderator guiding discussions while another observed non-verbal cues. Qualitative content analysis revealed key barriers across four dimensions: technological (interface complexity, infrastructure limitations), employee (professional identity conflicts, interpersonal tensions), task-related (real-time monitoring challenges, reduced pedagogical flexibility), and organizational (inadequate support systems, unclear implementation policies). These findings suggest that successful AI integration in PE requires a holistic approach addressing both technological and human factors, rather than focusing solely on technological advancement. The study contributes to understanding how socio-technical interactions uniquely manifest in physically active learning environments.

## 1. Introduction

The rapid advancement of artificial intelligence (AI) has garnered significant attention in educational circles, with AI widely regarded as a transformative tool capable of enhancing teaching efficiency and improving learning outcomes ([16]; [23]). In particular, AI’s potential to revolutionize classroom instruction has been emphasized, offering a range of benefits such as personalized learning, intelligent classroom management, and data-driven insights ([37]). Physical education (PE), as an integral part of the education system, has not been excluded from this trend, with AI seen as a promising solution to improve classroom dynamics, customize training programs, and increase student engagement ([45]).

In recent years, the integration of artificial intelligence (AI) into physical education (PE) has garnered considerable momentum. Studies indicate that AI is transforming conventional PE pedagogy, particularly by enhancing personalized instruction, fine-grained motion analysis, and objective performance assessment. Through an extensive review covering two decades of research on AI in education, [7] ([7]) delineated three core areas in which AI is adopted for PE: teaching assistance systems, analytics-based assessment, and intelligent learning environments. Notably, in the aftermath of the COVID-19 pandemic, [30] ([30]) proposed a virtual PE classroom paradigm that broadens possibilities for remote instruction, leveraging advanced algorithms for real-time evaluation of students’ athletic activities and alleviating spatial constraints. In discipline-specific applications, AI-driven innovations have shown substantial promise. For example, [31] ([31]) introduced an intelligent assistance system underpinned by high-order complex network theory, enabling precise tracking of athletes’ movements and delivery of individualized corrective feedback. Likewise, [21] ([21]) applied deep learning methods to football instruction, producing data-driven insights for coaching decisions. Meanwhile, [35] ([35]) combined virtual reality with particle swarm optimization to improve collegiate PE coursework, reporting notable gains in student engagement and learning outcomes. Beyond instruction, AI-powered tools for monitoring and evaluation have also proven effective. [21] ([21]) designed a fuzzy evaluation framework that relies on AI to assess various PE teaching methods, whereas [14] ([14]) formulated an AI-centered sports management system capable of systematically tracking and analyzing students’ performance data. Such advances contribute substantially to enhancing both objectivity and efficiency in PE assessments. Looking ahead, as wearable technologies and virtual reality continue to evolve, AI applications in PE are anticipated to grow even more sophisticated. [19] ([19]) predict that intelligent, student-centered PE models will emerge as the new standard. These innovations stand to elevate the quality of PE instruction while enriching students’ overall learning experiences. Consequently, uncovering effective mechanisms for AI integration and exploring next-generation deployment strategies remain vital areas for forthcoming research.

However, despite the widespread enthusiasm surrounding AI’s potential in education, its application in PE classrooms remains complex and fraught with challenges. The dynamic, interactive, and physically intensive nature of PE instruction sets it apart from other academic disciplines, creating a unique environment in which AI integration may not always yield the anticipated benefits ([1]; [40]). While existing research has focused primarily on the technological aspects of AI—such as algorithmic design and system optimization ([19]; [43])—there is a notable gap in the literature regarding the subjective experiences of PE teachers. Few studies have explored how AI impacts the teaching process from the perspective of educators, particularly in terms of its practical effectiveness and the challenges it may introduce into the classroom. This gap underscores the need for research that not only examines the potential benefits of AI in PE but also critically investigates how the technology interacts with the unique characteristics of the subject and the subjective experiences of teachers, with an awareness that AI may, in some cases, introduce unintended challenges or drawbacks.

## 2. Framework of Social Technology System Theory

Socio-technical systems (STS) theory originated from studies conducted by the Tavistock Institute in the early 1950s to understand how technological changes affected social relationships in industrial settings ([33]). Initial work investigated the introduction of mechanized coal-mining methods, revealing that purely technical improvements could fail if they neglected workers’ social and psychological needs. Through subsequent refinements, scholars such as [12] ([12]) and [8] ([8]) emphasized that joint optimization of the “social system” (e.g., human roles, relationships, and culture) and the “technical system” (e.g., machinery, processes, and tools) is critical to achieving organizational effectiveness. Over time, STS theory evolved from its industrial roots to incorporate broader concerns like ergonomics, participatory design, and policy formation. This evolution has highlighted how interdependencies between social and technical components are dynamic, emergent, and shaped by context. Essentially, the core premise of STS theory is that technology adoption cannot be divorced from the people, structures, and cultures that surround it, and any attempt to optimize one subsystem while ignoring the other may inadvertently lead to inefficiency or resistance.

Within educational research, STS theory has attracted interest as digital tools and platforms increasingly permeate classrooms and curricula ([9]; [26]). Investigations in this area suggest that introducing new technologies—such as learning management systems, mobile devices, or AI-driven tutoring programs—often requires alignment with institutional resources, pedagogical beliefs, and student engagement practices ([32]). For example, teachers who feel inadequately supported may resist or superficially implement digital tools, thus diminishing their potential benefits. Likewise, STS-informed studies have analyzed the ways in which social factors (e.g., professional development opportunities, peer collaboration) and technical elements (e.g., software usability, hardware availability) intersect in shaping teacher adoption behaviors. By accounting for the interplay between educational stakeholders and emergent technological affordances, STS research in education underscores the need for holistic strategies that address infrastructural, professional, and cultural dimensions simultaneously.

As artificial intelligence applications gain momentum across sectors, researchers have increasingly turned to STS theory to explore challenges and impacts associated with AI implementation ([27]; [20]). Particularly in AI-rich contexts, systems are not merely tools but active participants that interact with social structures, such as organizational hierarchies and professional norms ([24]). Studies applying STS frameworks have shown how AI algorithms can reconfigure work routines, reshape user roles, and prompt new ethical concerns ([3]). In education, for instance, AI-driven analytics may provide rapid feedback on student performance or recommend individualized learning paths. However, if educators lack the training, time, or trust to integrate AI insights into their daily practices, these innovations may yield more confusion than clarity ([41]). By examining AI deployment as a socio-technical phenomenon, scholars can identify the interdependencies among technical aspects (e.g., data modeling, interface design) and social elements (e.g., teacher expertise, institutional policies) that contribute to AI’s success or failure.

Building on these insights, the current research applies STS theory to investigate why, in the domain of primary school physical education, AI may paradoxically hinder rather than improve teaching efficiency. Primary school PE involves highly interactive, real-time dynamics, where tasks cannot be easily standardized and require instructors’ continuous adaptation. Applying an STS lens captures the interplay of social factors—like teachers’ pedagogical goals, institutional support, and student engagement—with AI technologies that offer data-driven analytics or generative lesson suggestions ([43]). To systematically assess these interdependencies, this study constructs a four-dimensional conceptual framework encompassing Technology, Task, Teacher, and Organization. The framework examines how AI tools align with PE tasks, how teachers adopt and trust new functionalities, and how school-level policies or resource constraints facilitate or impede meaningful integration. Through this perspective, the research aims to provide a nuanced understanding of both the promises and pitfalls of AI-driven solutions, offering strategic insights for policymakers, developers, and educators seeking to harmonize technical capabilities with the social realities of PE instruction.

## 3. Research Question

Despite the optimistic depictions of AI’s potential to enhance physical education (PE) teaching, existing literature often focuses disproportionately on the technical features of AI technologies while overlooking the subjective experiences and social factors that influence teachers’ use of these tools. This oversight leaves a gap in understanding why, in practical settings, AI might not always empower teachers or even hinder teaching effectiveness. More specifically, the challenge lies in explaining how social contexts, teacher attitudes, and interaction dynamics may mitigate the intended benefits of AI integration. Building on socio-technical systems theory, this study aims to address this gap by exploring the factors that contribute to the complex relationship between AI technologies and PE teaching practices. The research focuses on two core questions:

What are the key factors at the technological, task, teacher, and organizational levels that prevent AI technologies from effectively empowering PE teaching?

How do these factors interact to form a set of barriers that hinder the effective application of AI in PE teaching?

Through focus group interviews, this study aims to examine how AI technologies can both enhance and potentially impede teaching efficiency in physical education.

## 4. Materials and Methods

### 4.1. The Application of Phenomenological Research Design

This study employed a phenomenological research design to examine the complex interplay between artificial intelligence (AI) technologies and physical education (PE) practices. The phenomenological approach was selected as the methodological framework for its capacity to illuminate participants’ lived experiences and shared meanings regarding AI implementation challenges ([25]; [34]). While quantitative methodologies predominantly rely on numerical data and predefined variables, phenomenological inquiry enables a nuanced understanding of educators’ subjective experiences and perceptions ([10]; [2]; [39]). Contemporary research on AI integration in PE settings has largely emphasized technological functionality, often reducing it to a mechanical intervention while overlooking the nuanced experiential dimensions of frontline educators. Our phenomenological framework facilitated exploration beyond surface-level observations to examine the underlying structures of meaning in teachers’ encounters with AI technology. To gather rich phenomenological data, we utilized focus group interviews as the primary data collection method. This approach proved particularly efficacious in capturing multiple perspectives simultaneously while fostering dynamic discussions that revealed shared experiences and collective meaning-making processes. The methodological choice of focus groups has demonstrated considerable utility in educational research contexts ([38]), specifically in investigating teachers’ engagement with digital technologies and their perspectives on pedagogical integration ([6]; [13]). The synergistic nature of focus group interactions enabled participants to articulate and reflect upon their experiences while facilitating the emergence of deeper insights into the complexities of AI implementation in PE settings ([18]).

### 4.2. Study Design and Setting

#### 4.2.1. The Practical Background of the Research

This study is situated within a larger project focused on the integration of artificial intelligence (AI) into primary school physical education (PE) teaching. The initial aim of the project was to enhance teachers’ instructional practices by training them to effectively use AI technologies in their classrooms, with the expectation that such integration would lead to improved teaching quality and student engagement. However, as the project progressed, a noteworthy pattern emerged in the feedback provided by a segment of participating teachers. Specifically, some teachers reported that once the project concluded, they reverted to traditional PE teaching methods, no longer utilizing the AI tools introduced during the intervention. This return to conventional practices prompted a deeper inquiry into the reasons behind this disengagement. Further probing revealed that these teachers felt that AI had not, in fact, empowered or significantly improved their teaching. Their assessments were based on subjective judgments, reflecting a perceived mismatch between the potential of AI and its actual impact on their pedagogical effectiveness. This discrepancy—between the initial expectations of AI’s role and its actual impact on teaching practices—raised important questions about the contextual factors and underlying reasons why AI may not always succeed in transforming classroom practices. It is this gap that the current study seeks to address by exploring the factors that hinder AI’s effectiveness in primary school PE teaching and examining the broader implications of these findings for educational technology integration.

#### 4.2.2. Sampling Strategy

This study is part of a larger research project aimed at investigating the effectiveness of physical education (PE) teaching and the potential role of technological integration in enhancing classroom practices. The project was conducted in two cities, Nanjing and Chongqing, each representing distinct educational contexts within China. In both cities, we selected qualified primary school PE teachers from several core schools—each with a student body of over 500 and having worked exclusively on AI teaching for at least the past two years. These schools were chosen for their active involvement in integrating AI technologies into their PE classrooms, providing a unique opportunity to explore the practical impact of AI on teaching practices.

To ensure the study aligned with its objectives, a purposeful sampling strategy was employed to select participants ([29]). Specifically, teachers were invited to complete an open-ended questionnaire designed to assess their subjective evaluations of AI’s effectiveness in the classroom. Based on their responses, teachers who expressed more negative evaluations—those who believed that AI had not significantly enhanced or empowered their teaching—were selected for follow-up focus group interviews. This purposeful sampling approach ensured that the study focused on teachers who had direct experience with AI technologies but had not found them effective in improving their teaching practices, thus aligning with the research goal of exploring why AI may not always foster educational enhancement. Further, to extend the sample and ensure it reflected a broad range of experiences, snowball sampling was employed ([4]). Teachers who expressed dissatisfaction with the AI integration were asked to recommend other colleagues with similar views or experiences. This allowed for additional participants to be recruited through professional networks, ensuring that the sample continued to represent teachers who had actively engaged with AI in PE teaching and were able to critically assess its impact.

The decision to focus specifically on primary school PE teachers was driven by several key considerations. First, primary school PE teaching emphasizes fun, engagement, and game-based learning, which are particularly distinct from other academic subjects. This made the integration of AI in primary school PE classrooms a unique case for investigation, as the interaction between AI technologies and these playful pedagogical elements provided an opportunity to investigate a complex and underexplored dynamic. Second, focusing on primary school teachers allowed for a deeper, context-specific exploration of the challenges and opportunities associated with AI integration, enabling the study to generate more targeted insights into this specific educational level. The focus group interviews were conducted in neutral, comfortable settings within the schools, such as meeting rooms, to foster open and candid discussions. In line with ethical standards and privacy considerations, no video recordings were made; instead, audio recordings were used, with transcription and analysis conducted thereafter. Upon arrival, each participant signed an informed consent form detailing the study’s purpose, procedures, and confidentiality measures. The interviews employed a dual-facilitator approach: one facilitator (the primary author) guided the discussion and interview questions, while the second facilitator observed participants’ non-verbal cues (such as body language and facial expressions). The second facilitator provided real-time suggestions to the lead facilitator, ensuring that further probes or clarifications could be introduced based on the participants’ reactions. Specific sample information is shown in Table 1.

#### 4.2.3. The Data Collection Tool

The design of the data collection tool for this study was informed by recent literature on AI integration in education, specifically in physical education (PE). Drawing from studies like [19] ([19]), who examined the technical challenges of AI tools in PE, the survey includes questions on issues like software complexity and compatibility with existing teaching systems. Their findings suggested that technical difficulties such as complex interfaces can hinder effective AI adoption, which influenced questions such as, “What specific technical difficulties have you encountered when using AI tools?” The Teacher Dimension was shaped by [36] ([36]), who highlighted the importance of teacher attitudes in adopting AI. Their research pointed to the significance of teachers’ perceptions of AI’s usefulness and ease of use in determining its success. Based on this, the survey includes questions on teachers’ attitudes towards AI in PE and how these shape team dynamics and feedback, such as, “What are your and your colleagues’ overall attitudes toward the introduction of AI in PE?” The Tasks Dimension was developed with insights from [45] ([45]), who identified areas where AI struggles to support certain teaching tasks, particularly those requiring personal interaction or spontaneous decisions. This led to questions about which tasks AI performs poorly in supporting, for example, “In which teaching tasks do you believe AI technologies perform poorly?” Finally, the Organizational Structure Dimension reflects the barriers identified by [5] ([5]) and [28] ([28]), who found that organizational culture, lack of resources, and unclear policies are significant barriers to effective technology integration. This informed questions about the role of school policies and resource availability in AI adoption, such as, “Do you believe the school lacks sufficient resources and support in promoting AI technologies?”

Overall, the design of the open-ended questions was grounded in these studies, ensuring that the tool captures the key challenges and factors influencing AI adoption in PE classrooms. To ensure the scientific rigor and validity of the data collection tool, its development was closely guided by three experts with doctoral degrees in Educational Technology. The process began by reaching out to these experts via email, where the initial draft of the survey was shared, accompanied by a detailed explanation of the study’s aims and objectives. The experts provided written feedback on the clarity of the questions, the relevance of the dimensions being measured, and the alignment with the theoretical framework.

Furthermore, to facilitate a deeper discussion, a formal expert review meeting was organized, in which all three experts participated. During this session, the experts provided additional insights into the content and structure of the survey, focusing on how well the tool captured the complexities of AI integration in physical education. Based on their feedback, the survey was revised to improve its comprehensiveness, precision, and ability to capture the nuances of teachers’ experiences with AI technologies. This expert consultation process was critical in ensuring the methodological soundness of the measurement tool and enhancing its validity for this specific research context.

### 4.3. Procedures, Data Collection, and Analysis

The focus group protocol was developed collaboratively by a research team comprising experts from diverse academic backgrounds: a physical education teacher education researcher, an information technology researcher, and a teacher education researcher. The protocol development process involved extensive consultation with physical education department heads from two participating schools to ensure practical relevance and pedagogical appropriateness.

Participants were selected through a purposive sampling strategy based on specific criteria: (1) a minimum of three years of physical education teaching experience; (2) current involvement in technology-integrated PE instruction; and (3) participation in at least one AI-related professional development program within the past two years. Detailed demographic information was collected through preliminary screening questionnaires to ensure participant suitability and enhance research reliability. Two face-to-face focus groups were conducted in different geographical locations: one in Nanjing (*n* = 6) and one in Chongqing (*n* = 7). The sessions took place in designated meeting rooms at participating schools, allowing for direct observation of non-verbal cues and facilitating natural group dynamics. Secondary moderators were specifically trained to observe and document participants’ non-verbal cues, such as hesitation, disagreement, or discomfort. When such cues were observed, moderators employed targeted follow-up questions (detailed in Table 2) to probe deeper into participants’ underlying thoughts and concerns. 

This geographical diversity was intentionally designed to capture potential regional variations in teaching experiences and challenges. The second and third authors served as secondary moderators for the Nanjing and Chongqing sessions, respectively, ensuring consistent documentation and facilitation support across both locations. Prior to the commencement of each focus group session, all participants were required to complete informed consent forms. The forms explicitly stated that participation was voluntary and that participants could withdraw from the discussion at any time without any negative consequences. This approach aligned with established ethical guidelines for qualitative research involving human subjects.

Each 90 min session began with an ice-breaking activity facilitated by the first author, who served as the primary moderator. The initial discussion centered around the broader theme of “effective physical education teaching strategies,” which was deliberately chosen to create a comfortable and engaging atmosphere for participant interaction. This preliminary discussion lasted approximately 15 min and successfully established rapport among participants before transitioning to the main focus group discussion.

To minimize potential peer influence and ensure balanced participation, several strategies were implemented: (1) structured turn-taking techniques ensuring equal participation opportunities; (2) five-minute independent reflection periods before each discussion topic; (3) proactive solicitation of diverse viewpoints through prompts such as “Does anyone have a different perspective?”; (4) a “round-robin” approach for key questions; and (5) secondary moderator monitoring of group dynamics with written cues to the primary moderator when participants appeared hesitant to contribute.

Following the ice-breaking session, the primary moderator guided participants through a structured discussion focusing on their experiences and challenges with implementing artificial intelligence in physical education instruction. The discussion prompts were designed to elicit detailed responses about both the practical difficulties encountered and the potential underlying causes of these challenges (the specific discussion prompts are detailed in Table 2).

In analyzing the data from the focus group discussions, we employed qualitative content analysis, a flexible and dynamic approach that allowed us to capture the complexity and depth of the teachers’ perspectives. To establish trustworthiness, we implemented multiple validation strategies. Three independent experts in physical education and educational technology reviewed our coding scheme and thematic analysis, providing valuable feedback that led to refinements in our analytical categories. Following initial analysis, we conducted member checking sessions with selected participants to verify the accuracy of our interpretations. The analysis proceeded through several stages. Initially, all focus group transcripts were independently read by the research team members, with each researcher coding the data based on emerging themes. Following this, the team met to compare and refine the codes, discussing discrepancies and ensuring that the coding process remained aligned with the research objectives. This iterative process, combined with expert validation and member checking, strengthened the validity of the findings. The research team maintained a comprehensive audit trail documenting all methodological decisions and analytical insights. The final step involved the categorization of codes into broader themes that reflected the key issues impacting teachers’ perceptions of AI integration in physical education classrooms. By employing this collaborative and iterative coding process, enhanced by multiple validation strategies, we ensured a robust analysis that accurately represented the complexities of the teachers’ experiences with AI technology. Crucially, the coding process was deeply informed by the socio-technical systems Theory, which guided the identification of factors impacting the interaction between technology and teaching practices. This theoretical framework helped ensure that both technological and pedagogical aspects were equally emphasized in the analysis, offering a holistic understanding of the challenges faced by PE teachers.

## 5. Results

The analysis of teacher interviews uncovered a multifaceted array of barriers that hinder the effective integration of artificial intelligence (AI) in physical education (PE) instruction. These barriers were systematically categorized into four primary dimensions—technology, employee, tasks, and organizational structures—guided by the socio-technical systems (STS) framework. Each dimension encapsulates specific challenges that collectively impede AI’s potential to enhance teaching efficiency and instructional quality. The ensuing sections delineate these barriers in detail, supported by representative examples from teachers’ experiences, thereby illustrating the complex interplay of factors that limit AI’s transformative impact in PE settings.

### 5.1. Results of Technology Dimension

Analysis of the technological dimension revealed three interconnected themes that systematically impede AI integration in physical education instruction. These themes emerged through rigorous coding of interview data, reflecting the complex interplay between technological infrastructure, pedagogical needs, and institutional support systems. The specific coding results are shown in Table 3.

#### 5.1.1. Value and Relevance

The primary theme encompasses fundamental concerns about AI’s pedagogical value proposition in PE contexts. Interview data revealed persistent skepticism about AI’s capacity to enhance traditional teaching methodologies. As articulated by a veteran teacher with eight years of experience: “Despite the promised benefits, I’ve yet to see concrete evidence that these AI tools meaningfully improve our established teaching methods” (NJ002). This sentiment was substantiated by another participant who elaborated on the misalignment between AI capabilities and instructional requirements: “The sophisticated analytics offered by AI systems often miss the mark in addressing our core pedagogical needs, particularly in dynamic PE environments where immediate, context-aware responses are essential” (CQ004). These perspectives underscore a critical gap between technological sophistication and practical utility.

#### 5.1.2. Operational and Infrastructural Burdens

The second theme illuminates the substantial operational challenges inherent in AI implementation. Systematic analysis of interview data revealed multifaceted technical barriers that significantly impact teaching efficiency. A department head with 12 years of experience described the interface-related challenges: “The complexity of the user interface creates a cognitive burden that diverts attention from our primary instructional responsibilities” (NJ005). Infrastructure limitations further compound these difficulties, as evidenced by one participant’s observation: “The system’s heavy reliance on network connectivity introduces a layer of unpredictability that fundamentally compromises our ability to maintain consistent instructional quality” (CQ002). These operational constraints are exacerbated by technical instabilities, as another teacher noted: “The frequent system updates and compatibility issues create persistent disruptions to our established teaching rhythms” (NJ001).

#### 5.1.3. Inadequate Support and Preparedness

The final theme reveals systematic deficiencies in technical support infrastructure and professional development frameworks. Interview data highlighted a critical gap between initial implementation and sustained support mechanisms. A teacher with doctoral-level expertise in physical education emphasized: “The absence of comprehensive, ongoing technical training significantly undermines our capacity to leverage AI’s potential effectively” (CQ006). This structural inadequacy is particularly evident in crisis management scenarios, as detailed by another participant: “The delayed response to technical issues creates cascading disruptions that compromise not just individual lessons but entire units of instruction” (NJ004).

### 5.2. Results of Teacher Dimension

Our analysis of the teacher dimension illuminated a complex tapestry of interpersonal and organizational dynamics that significantly influence AI integration in physical education contexts. Through rigorous examination of interview data, four distinctive yet interwoven themes emerged, each revealing unique facets of the social challenges inherent in technological transformation. The specific coding results are shown in Table 4.

#### 5.2.1. Negative Attitudes and Lack of Trust

The pervasive skepticism toward AI integration manifests not merely as individual resistance but as a deeply embedded collective sentiment within the teaching community. This phenomenon was eloquently captured by a veteran educator: “The reluctance we’re witnessing isn’t simply technological hesitation—it reflects a fundamental concern about preserving the essence of physical education” (NJ003). Such perspectives, emerging consistently across interviews, suggest that resistance to AI extends beyond mere technological skepticism to encompass broader concerns about pedagogical identity and professional autonomy. Another participant’s reflection proved particularly illuminating: “Our department’s apprehension stems from a shared commitment to maintaining the human connection that defines effective PE instruction” (CQ003).

#### 5.2.2. Collaboration and Conflict

The implementation of AI technologies has catalyzed notable shifts in professional relationships and departmental dynamics. In-depth analysis revealed how technological adoption creates subtle yet significant divisions within teaching teams. A particularly insightful observation came from a department head: “The introduction of AI has inadvertently created invisible barriers between early adopters and those who prefer traditional methods” (CQ002). These divisions manifest in diminished knowledge sharing and collaborative practices, as evidenced by one participant’s astute observation: “What’s particularly concerning is how technological differences have begun to erode our previously strong culture of peer support and mentorship” (NJ005).

#### 5.2.3. Leadership and Acceptance Variability

The data revealed a nuanced interplay between leadership approaches and teacher receptivity to AI integration. Rather than simple binary responses, the analysis uncovered a spectrum of acceptance patterns influenced by leadership signals. A participant with extensive administrative experience noted: “Leadership’s ambivalence about AI’s role creates ripple effects throughout our department, affecting everything from resource allocation to professional development priorities” (NJ005). This observation gained additional depth through another teacher’s insight: “The varying levels of engagement with AI reflect broader uncertainties about its alignment with our educational mission” (CQ004).

#### 5.2.4. Pressure and Competition

Perhaps most striking was the emergence of new competitive dynamics triggered by AI implementation. Beyond mere technological adoption, these pressures reshape professional relationships and departmental culture in unexpected ways. As articulated by one experienced educator: “The emphasis on AI proficiency has introduced subtle yet powerful competitive elements into what was previously a collaborative teaching environment” (NJ001). This transformation of professional dynamics carries significant implications for departmental cohesion, as highlighted by another participant: “The quantification of AI adoption success has inadvertently created hierarchies that undermine our collective educational mission” (CQ005).

### 5.3. Results of Tasks Dimension

Our investigation into the task dimension unveiled a fascinating dialectic between technological advancement and pedagogical authenticity in physical education. Through meticulous analysis of interview data, two fundamental tensions emerged, each revealing distinct challenges in reconciling AI integration with the essence of PE instruction. The specific coding results are shown in Table 5.

#### 5.3.1. Dynamic Environment Management

The inherent complexity of physical education environments presents unique challenges for AI integration, manifesting in ways that challenge conventional assumptions about technological enhancement. Within these dynamic spaces, teachers navigate an intricate balance between traditional pedagogical responsibilities and emerging technological demands. As articulated by a seasoned educator with extensive technological experience: “The art of PE instruction lies in our ability to read and respond to student movement in real-time. AI monitoring, while promising, introduces a parallel cognitive load that fragments our attention in crucial moments” (NJ004). This cognitive division becomes particularly pronounced in student engagement dynamics. A department coordinator offered a nuanced perspective: “The introduction of technological elements fundamentally alters the kinesthetic focus we strive to maintain. Students’ natural curiosity about devices creates a persistent tension between digital engagement and physical presence” (CQ001). Such observations illuminate how technological integration, despite its intended benefits, can inadvertently compromise the core objectives of physical education.

#### 5.3.2. Personalization Constraints

In a striking paradox, the implementation of AI-driven personalization features often results in decreased instructional adaptability. This contradiction emerges most visibly in the spatial and structural demands imposed by AI systems. A particularly illuminating insight came from a teacher with advanced pedagogical training: “The promise of personalized learning collides with the physical reality of our teaching spaces. AI’s rigid spatial requirements often constrain the very dynamism that makes PE instruction effective” (CQ005). Beyond physical constraints, the data revealed deeper tensions between algorithmic prescription and pedagogical intuition. As one participant with dual expertise in technology and physical education observed: “While AI systems offer sophisticated customization options, they inadvertently create instructional boundaries that limit our ability to respond organically to student needs” (NJ001). This observation gains additional resonance through another teacher’s reflection: “The spontaneity that often catalyzes breakthrough moments in physical learning becomes increasingly difficult to achieve within AI’s structured frameworks” (CQ003). The cumulative impact of these constraints manifests in what might be termed a “pedagogical paradox”—where tools designed to enhance personalization actually restrict the natural adaptability essential to effective PE instruction. This phenomenon emerges consistently across interviews, suggesting a fundamental misalignment between AI’s algorithmic approach and the organic, fluid nature of physical education pedagogy.

### 5.4. Results of Organizational Structure Dimension

Our investigation into the organizational dimension yielded rich insights into the institutional dynamics shaping AI integration in physical education contexts. Through careful analysis of interview data, we discovered a tapestry of interconnected challenges that collectively illuminate the complex relationship between organizational structures and technological innovation. The specific coding results are shown in Table 6.

#### 5.4.1. Administrative Pressures and Culture

The intersection of administrative imperatives and institutional culture emerged as a critical locus of tension in AI implementation efforts. Within this domain, we observed what might be characterized as an “implementation dialectic”—where administrative mandates generate their own resistance through the very mechanisms intended to ensure adoption. As one senior educator expressed: “We’re constantly being told to use these AI tools, but there’s no real support system in place. It feels like we’re being set up to fail” (NJ001). This dynamic finds particular resonance in the fluid nature of policy directives, with another participant noting: “Just when we think we’ve figured out what’s expected of us, the guidelines change again. It’s exhausting and frustrating” (CQ002).

#### 5.4.2. Resource and Support Deficiencies

Analysis revealed a sophisticated interplay between institutional aspirations and operational realities, manifesting in what we term a “capability-intention gap”. This phenomenon extends beyond mere resource constraints to encompass fundamental misalignments in organizational understanding and support structures. A veteran educator shared: “Our school leaders talk about AI transformation, but they don’t really understand what we need in the gym. The resources they provide often miss the mark completely” (NJ003). The reverberations of this misalignment echo through departmental boundaries, as one participant observed: “Every department is trying to figure this out on their own. We barely know what other PE teachers in our own school are doing, let alone other departments” (CQ004).

#### 5.4.3. Professional Autonomy Limitations

The data illuminated a nuanced pattern of professional marginalization that extends beyond simple exclusion from decision-making processes. One participant candidly expressed: “Nobody asked us what tools would actually help in our PE classes. They just bought these expensive systems and now expect us to make them work somehow” (NJ005). This observation gains particular salience through another educator’s experience: “We’ve been teaching PE for years; we know what works with our students. But suddenly, all that experience seems to count for nothing” (CQ006).

#### 5.4.4. Strategic Planning Gaps

Perhaps most compelling was the emergence of what we identify as “strategic discontinuity”—a phenomenon characterized by the absence of cohesive long-term planning frameworks. An educator with extensive experience shared: “It feels like we’re just jumping from one AI initiative to another without any real plan. There’s no clear picture of where we’re heading with all this” (CQ001). This sentiment was echoed by another participant: “We have no way to know if these AI tools are actually improving our teaching because nobody’s really tracking what works and what doesn’t” (NJ004).

### 5.5. Results of Examples Provided by Teachers

To protect participant privacy while illustrating the intersecting challenges documented across multiple participants, we combined elements from several teachers’ accounts into a single composite narrative, represented here through “Ms. X” (a pseudonym).

Ms. X’s attempt to integrate an AI video analysis tool into her PE lessons exemplifies how, despite initial enthusiasm, technological complexity and limited value-add quickly undermine its benefits. Frequent updates, intricate interfaces, and insufficient technical support consumed more of her time than anticipated. Meanwhile, colleagues questioned her professional integrity, viewing her reliance on AI-generated reports as “taking shortcuts” rather than recognizing the tool’s potential for refined assessment. In a dynamic PE setting, the equipment’s novelty distracted students, complicating classroom management, while the tool’s rigid spatial and activity requirements restricted Ms. X’s ability to adapt tasks spontaneously. At the organizational level, vague policies, a lack of coherent guidance, and minimal leadership support left her uncertain about long-term goals or appropriate uses of AI. In the absence of adequate training, strategic direction, or peer support, Ms. X’s foray into AI remained isolated and fraught with difficulties.

This composite scenario, drawn from multiple teacher narratives, encapsulates the multifaceted issues—technological, interpersonal, task-related, and organizational—that collectively hinder the effective application of AI in PE instruction.

## 6. Discussion

This investigation elucidates the intricate interplay of technological, human, pedagogical, and organizational dimensions that collectively shape artificial intelligence integration within physical education contexts. Through the analytical lens of socio-technical systems theory, our findings reveal that impediments to effective AI implementation transcend mere technical constraints, encompassing deeply embedded social and organizational dynamics that fundamentally influence technology adoption trajectories.

In examining the technological dimension, our findings both corroborate and extend extant research on educational technology adoption. While prior scholarship has predominantly emphasized universal barriers such as perceived ease of use and technical reliability, our empirical results illuminate PE-specific complexities, particularly regarding the simultaneous management of real-time data monitoring and immediate physical supervision. This fundamental tension between technological engagement and pedagogical presence aligns with [45]’s ([45]) findings about the challenges of integrating AI in dynamic physical education environments, while also resonating with [42]’s ([42]) observations concerning the implementation complexities of AI in specialized educational domains.

Our findings regarding teacher attitudes and professional dynamics substantively augment [32]’s ([32]) work while highlighting PE-specific concerns. The emergence of what we conceptualize as “pedagogical identity conflict”—wherein educators perceive AI as potentially undermining their professional autonomy in physical instruction—builds upon [19]’s ([19]) discourse on AI integration challenges in PE settings. This phenomenon transcends mere technological skepticism to encompass fundamental ontological questions about physical education’s nature, echoing [45]’s ([45]) concerns regarding the potential displacement of established PE pedagogical approaches while also reflecting [22] ([22]), emphasis on the need to preserve pedagogical flexibility in AI-enhanced environments.

Perhaps most significantly, our analysis reveals how PE instruction’s inherent characteristics engender distinct challenges for AI integration that may be absent in other academic disciplines. The identified tension between AI’s structured methodological approaches and PE’s requirement for pedagogical flexibility aligns with [31]’s ([31]) observations regarding current AI systems’ limitations in physical education contexts. This “rigidity-adaptability paradox” suggests that technologies designed to enhance instructional efficiency may inadvertently constrain the pedagogical spontaneity and responsiveness that [17] ([17]) identifies as fundamental to effective PE instruction.

The organizational barriers identified complement extant research on institutional change ([15]; [44]), while illuminating distinctive aspects of PE departments’ technological transformation. Our findings substantiate [40]’s ([40]) argument that conventional approaches to technology implementation, which frequently prioritize administrative efficiency over pedagogical effectiveness, may be particularly problematic in PE contexts. The resource and support deficiencies identified resonate with [21]’s ([21]) observations regarding the challenges of implementing AI-driven assessment systems without adequate institutional infrastructure, while also reflecting the systematic implementation challenges identified by both review papers ([45]).

These findings make several substantive theoretical contributions to our understanding of technology integration in specialized educational contexts. Foremost, they extend socio-technical systems theory by delineating specific mechanisms through which technical and social factors interact in physically active learning environments, building upon [33]’s ([33]) foundational work. Additionally, they introduce the concept of “embodied pedagogy resistance”—the unique challenges arising from attempts to digitize inherently physical educational practices. This theoretical advancement responds to [24]’s ([24]) call for a more nuanced understanding of socio-technical interactions in specialized contexts, while also addressing [11] ([11]) emphasis on the need for theoretical frameworks that can accommodate the unique characteristics of physical education. Based on these findings, we propose an integrated approach to address implementation challenges and advance future research in AI-PE integration. To overcome negative attitudes, institutions should establish clear pedagogical-technical alignment frameworks that demonstrate AI’s role in enhancing rather than replacing traditional PE practices while developing comprehensive professional development programs that address both technical skills and pedagogical concerns. This should be coupled with collaborative implementation networks where PE teachers can collectively shape AI integration practices in their specific contexts ([45]). Future research should prioritize longitudinal studies examining attitude evolution, comparative analyses of professional development approaches, and design-based research developing PE-specific AI tools. These initiatives should employ mixed-method approaches to capture both quantitative impacts and qualitative experiences, with particular attention to maintaining pedagogical flexibility while leveraging AI’s capabilities for enhancing physical education outcomes. Additionally, cross-cultural studies examining how different educational contexts influence AI integration could provide valuable insights for developing more nuanced implementation strategies, ultimately working toward realizing AI’s potential benefits while preserving the essential characteristics of physical education pedagogy. Based on the findings, we propose three key recommendations.

Establish a comprehensive PE-specific AI infrastructure assessment framework that systematically evaluates the compatibility between AI tools and physical education requirements. This framework should address three critical dimensions: (a) real-time monitoring capabilities that accommodate dynamic movement patterns in PE classes; (b) spatial flexibility that allows for diverse teaching arrangements and activities; and (c) seamless integration with existing PE equipment and facilities. The framework should incorporate quantifiable metrics for assessing technological performance while maintaining pedagogical effectiveness.

Implement a “PE Teacher-AI Co-design” initiative that actively engages physical educators in the development lifecycle of AI tools. This initiative should encompass: (a) early-stage consultation during AI tool development to ensure alignment with practical teaching needs; (b) iterative testing and feedback cycles in authentic PE environments; and (c) collaborative refinement of AI features based on actual classroom experiences. This approach ensures that technological solutions effectively address pedagogical requirements while maintaining the essential characteristics of physical education instruction.

Develop an integrated support ecosystem that bridges the gap between technical expertise and pedagogical practice. This ecosystem should include: (a) specialized technical staff with an understanding of PE-specific challenges; (b) structured peer learning networks for sharing successful AI integration strategies; and (c) regular professional development sessions combining technical skill enhancement with pedagogical applications. The system should incorporate continuous evaluation mechanisms to ensure sustained effectiveness and relevance to PE teaching requirements.

## 7. Conclusions

This study elucidates the intricate interplay between technological affordances and pedagogical praxis in physical education through a socio-technical systems theoretical lens. Our empirical findings reveal that the efficacy of AI implementation is constrained by multifaceted barriers across technological, interpersonal, task-oriented, and organizational dimensions, culminating in what we term “pedagogical-technological dissonance”. This phenomenon manifests when AI tools, ostensibly designed to enhance instructional efficiency, inadvertently impede the dynamic, embodied nature of physical education pedagogy. The findings suggest that successful AI integration necessitates a holistic reconceptualization that transcends mere technological sophistication, encompassing both the socio-cultural ecosystem of educational institutions and the phenomenological aspects of physical instruction. This theoretical advancement contributes to understanding the complex dialectic between digital innovation and pedagogical authenticity in physically active learning environments while offering pragmatic insights for educational technology implementation.

## 8. Limitations of This Research

Despite the valuable insights provided by this study regarding AI integration challenges in PE education, several limitations should be acknowledged. First, while our research captured teachers’ perspectives through focus groups, it did not employ longitudinal observations to track how these challenges evolve over time as teachers gain more experience with AI tools. Additionally, the study’s focus on primary school settings in two Chinese cities, while providing rich contextual data, may not fully represent the diverse educational environments across different cultural and socio-economic contexts. Another significant limitation lies in our inability to quantify the actual impact of identified barriers on student learning outcomes, as the study primarily relied on teacher perceptions rather than measurable performance indicators. Furthermore, while the socio-technical systems framework provided valuable theoretical grounding, the study did not fully explore how different combinations of technological and social factors might interact to create varying levels of implementation success or failure. Future research would benefit from employing mixed-methods approaches that combine qualitative insights with quantitative measures of both teacher adaptation and student achievement, potentially incorporating psychometric tools to assess teacher technology acceptance more systematically. Such studies could also examine how different school leadership styles and organizational cultures might moderate the relationship between AI implementation and teaching effectiveness in physical education settings.

## Figures and Tables

**Table 1 behavsci-15-00240-t001:** Interviewee information coding.

Participant ID	Region	Gender	Years of Teaching Experience	Education Level
NJ001	Nanjing	Female	5	Bachelor’s Degree
NJ002	Nanjing	Male	8	Master’s Degree
NJ003	Nanjing	Female	10	Bachelor’s Degree
NJ004	Nanjing	Male	7	Master’s Degree
NJ005	Nanjing	Female	12	Doctorate Degree
NJ006	Nanjing	Male	6	Bachelor’s Degree
CQ001	Chongqing	Female	4	Bachelor’s Degree
CQ002	Chongqing	Male	9	Master’s Degree
CQ003	Chongqing	Female	11	Bachelor’s Degree
CQ004	Chongqing	Male	5	Master’s Degree
CQ005	Chongqing	Female	7	Bachelor’s Degree
CQ006	Chongqing	Male	10	Doctorate Degree
CQ007	Chongqing	Female	3	Bachelor’s Degree

**Table 2 behavsci-15-00240-t002:** Semi-structured interview outline and questions.

Dimension	Question ID	Question	Sub-Questions/Notes
Technology Dimension	T1	What specific technical difficulties have you encountered when using artificial intelligence tools?	For example, is the software interface complex or are the feature settings not intuitive?
	T2	Do the current AI technologies in physical education meet your actual needs?	Which features are missing or unsuitable, preventing effective support for your teaching activities?
	T3	Have you experienced any compatibility issues between AI technologies and existing teaching equipment or systems during your use of AI-assisted teaching?	How have these compatibility issues affected your teaching efficiency?
	T4	How would you rate the user-friendliness of the current AI tools?	For example, does learning and mastering these tools require excessive time and effort, thereby impacting your teaching schedule?
	T5	Do you believe the current development direction of AI technologies in physical education aligns with your actual needs?	If not, in what aspects do you hope future AI technologies will be improved?
Teacher Dimension	P1	What are your and your colleagues’ overall attitudes toward the introduction of artificial intelligence technologies in physical education?	For example, are you all supportive, skeptical, or opposed? Please elaborate.
	P2	How is the discussion and interaction atmosphere regarding the use of artificial intelligence technologies within your research team or department?	How does this atmosphere affect the adoption and application of AI technologies?
	P3	How are colleagues who use AI to assist in teaching typically evaluated and received feedback within the team?	Do these evaluations affect other teachers’ acceptance and willingness to use AI technologies?
	P4	What specific impacts has the introduction of artificial intelligence technologies had on the collaboration and working relationships within your team?	Are these impacts positive or negative? Please provide examples.
	P5	Are there any disagreements or conflicts within the team regarding the use of artificial intelligence technologies?	In what aspects do these disagreements manifest, and how do they affect the application of AI technologies in teaching?
Tasks Dimension	TA1	How have your teaching tasks specifically changed with the integration of artificial intelligence technologies?	How have these changes affected your daily teaching work? Do you believe these changes have increased your workload or complexity?
	TA2	During the use of AI-assisted teaching, have you found certain teaching tasks becoming more difficult to manage or execute?	Please describe which tasks have been negatively impacted and how these impacts manifest.
	TA3	In which teaching tasks do you believe AI technologies perform poorly in supporting or optimizing them?	Why have these tasks not been effectively supported by AI? Are there teaching aspects or content that AI cannot adequately handle?
	TA4	Has the integration of AI technologies caused any parts of your teaching process to become disjointed or uncoordinated?	How do these disjointed or uncoordinated aspects affect overall teaching efficiency and effectiveness?
	TA5	Have you encountered mismatches between your teaching tasks and the support provided by AI technologies?	For example, do the analyses or feedback provided by AI align with your teaching objectives or methods? How does this mismatch affect your teaching practice?
Organizational Structure Dimension	O1	Has the school enforced the use of artificial intelligence technologies through administrative pressure?	For example, are there explicit requirements from higher authorities to use AI tools or include AI usage in performance evaluations? What negative impacts has this administrative pressure had on your teaching arrangements and job satisfaction?
	O2	Do you believe the school lacks sufficient resources and support in promoting artificial intelligence technologies?	For example, are there issues like inadequate funding, lack of necessary equipment, or insufficient technical support teams? How exactly do these resource and support deficiencies hinder your effective use of AI technologies?
	O3	Are school policies regarding the use of artificial intelligence technologies unclear or subject to frequent changes, thereby affecting your use of AI technologies?	For example, are the AI usage-related policies vague or frequently adjusted during implementation? How does this policy uncertainty make it difficult for you to effectively apply AI technologies in teaching?
	O4	Does the school’s organizational culture hold a negative attitude towards the use of artificial intelligence technologies or lack support for them?	For example, does the school prefer traditional teaching methods and lack acceptance and encouragement for new technologies? How does this cultural atmosphere limit your and your colleagues’ enthusiasm for trying and applying AI technologies in teaching?
	O5	In the school’s decision-making process, are teachers’ opinions and feedback ignored, leading to ineffective application of AI technologies?	For example, have you had the opportunity to participate in the formulation of AI-related policies and technology selection? If not, how does the lack of teacher participation result in AI technologies not effectively meeting teaching needs?
Additional Questions	A1	Overall, what do you believe are the main reasons why AI technologies have not significantly enhanced your teaching efficiency?	Can you share some specific examples or experiences?
	A2	In your teaching practice, what specific situations or factors have prevented AI technologies from performing as expected?	What are these specific situations or factors?
	A3	What are your views on the future application of AI technologies in physical education?	How do you think AI can be improved or adjusted to more effectively enhance teaching efficiency?

Note: Sub-questions were used as necessary probing questions during the interviews to elicit more detailed responses and deeper insights from participants when initial answers required further exploration.

**Table 3 behavsci-15-00240-t003:** Technology Dimension coding results.

Categories	Codes	Explanation	Frequency
Value and Relevance	Lack of Technological Advantages	Teachers’ perception that AI tools don’t provide significant improvements over traditional methods in PE teaching	32
	Mismatched Functionality	Disconnect between AI features and actual PE teaching requirements	28
Operational and Infrastructural Burdens	Inconvenient User Interface	Poor interface design hampering effective use during PE classes	35
	High Cognitive Costs	Substantial mental effort required to learn and operate AI systems	30
	Frequent Software Updates	Continuous system updates disrupting teaching flow	21
	Device and System Incompatibility	Technical conflicts between AI tools and existing school infrastructure	25
	High Network Dependency	Over-reliance on stable internet connectivity in PE settings	38
Inadequate Support and Preparedness	Lack of Systematic Technical Training	Absence of structured programs for AI tool mastery	34
	Insufficient Technical Support	Delayed or inadequate assistance for technical issues	31

Note: Frequency indicates the number of times each code appeared across all interview transcripts. Categories and codes emerged through systematic qualitative content analysis of focus group data.

**Table 4 behavsci-15-00240-t004:** Teacher Dimension coding results.

Categories	Codes	Explanation	Frequency
Negative Attitudes and Lack of Trust	Overall Unsupportive Attitudes	General resistance and negative disposition towards AI integration in PE teaching	35
	Resistant Team Atmosphere	Collective skepticism and opposition to AI adoption in teaching practices	32
Collaboration and Conflict	Lack of Trust in AI Technologies	Fundamental doubts about AI’s reliability and effectiveness among teaching staff	30
	Negative Evaluation of Colleagues Using AI	Critical judgment and skepticism towards teachers who embrace AI tools	28
	Poor Impact of AI on Team Collaboration	Deterioration of teamwork and cooperative relationships due to AI implementation	26
	AI-Induced Team Disagreements	Conflicts and tensions arising from differing views on AI adoption	24
	Lack of Peer Technical Sharing	Insufficient knowledge sharing and mutual support in AI implementation	31
Leadership and Acceptance Variability	Team Leadership’s Reserved Attitude	Hesitant and cautious approach from leadership towards AI adoption	29
	Variability in Teachers’ Acceptance	Inconsistent levels of AI acceptance and adoption among teaching staff	27
Pressure and Competition	Increased Competition and Pressure	Enhanced workplace stress and rivalry due to AI implementation demands	33

**Table 5 behavsci-15-00240-t005:** Tasks Dimension coding results.

Categories	Codes	Explanation	Frequency
Dynamic Environment Management	Real-time Data Monitoring Challenges	Difficulty in monitoring AI feedback while maintaining student safety and movement quality in active PE settings	36
	Student Device Distractions	Management of student attention being diverted by AI tools and devices	33
Personalization Constraints	Complex Lesson Organization	Increased complexity in organizing and overseeing lessons due to simultaneous physical and digital requirements	31
	Spatial Requirements	Physical space limitations and layout restrictions imposed by AI technology implementation	29
	Limited Teaching Flexibility	Reduced ability to adapt teaching strategies and activities spontaneously	32
	Predefined Instructional Sequences	Rigid adherence to AI-determined lesson structures limiting teacher autonomy and creativity	28

**Table 6 behavsci-15-00240-t006:** Organizational Dimension coding results.

Categories	Codes	Explanation	Frequency
Administrative Pressures and Culture	Administrative Pressure to Mandate AI	Top-down mandates for AI adoption without adequate support	35
	Unclear School Policies	Ambiguous or frequently changing guidelines for AI implementation	32
Resource and Support Deficiencies	Negative Organizational Culture	Institutional resistance and skepticism towards AI integration	29
	Leadership’s Limited Understanding	Insufficient leadership comprehension of AI implementation needs	31
	Inadequate Resources and Support	Lack of necessary technical resources and assistance	36
	Limited Cross-Departmental Collaboration	Poor coordination between different school departments	28
	Unsystematic Technology Promotion	Lack of structured approach to AI implementation	30
	Insufficient Training Programs	Absence of continuous professional development for AI	33
Professional Autonomy Limitations	Ignored Teacher Feedback	Disregard for teachers’ input and experiences	34
	Limited Decision-Making Participation	Exclusion from AI-related policy and implementation decisions	31
	Restricted Technology Input	Minimal teacher involvement in AI tool selection	27
Strategic Planning Gaps	Ineffective Performance Evaluation	Poor assessment of AI usage in teaching evaluations	26
	Absent Long-Term Strategy	Lack of comprehensive planning for sustainable AI integration	32

## Data Availability

Considering that the interview data all involve the personal privacy of teachers, it is not disclosed, but the coded data can be obtained by applying to the corresponding author.

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
