# Peer review of "The Paradox of AI Empowerment in Primary School Physical Education: Why Technology May Hinder, Not Help, Teaching Efficiency"

_behavsci, 2025, doi:10.3390/bs15020240_

Round 1

Reviewer 1 Report

Comments and Suggestions for Authors

1. In the summary section, the method used in the analysis and sample selection should be mentioned. 

2. In the summary section, it should be stated in which age group the participants attended the lessons of the students.

3. The first paragraph in the introduction (lines 27-34) mentions the general use of artificial intelligence in education. However, the second paragraph (lines 35-44) is not sufficient to justify the research. Why the research is necessary should be explained in more detail. The fact that there are only a few studies on this topic in the literature cannot be a sufficient reason for writing this article. The topic should be supported theoretically. 

4. Have the additional costs to education expenditures of the artificial intelligence used to evaluate student performance in lines 72 and 73 been taken into account? Are these students professional athletes? For which grades are these assessments required? Should these assessments be conducted in professional sports schools? Should they be conducted in public schools across the country? Aren't the technologies you mentioned already in current use in gyms? I think a more detailed explanation including these questions along with the existing explanations in the article would be appropriate. You need to provide theoretical support for the topic. 

5. 5. I came to the conclusion that the introduction was written in a single paragraph and then divided into chapters with headings. Each heading should be discussed separately and in more detail. When starting a paragraph, do not start sentences using “conjunctions” as in line 89.

6. The conceptual section needs more literature support. 

7. It is more appropriate to use the methods section to talk about the benefits of qualitative research. From the reader's point of view, it may prevent a holistic view of the conceptual framework in the introduction. 

8. A citation is required for lines 147-149. 

9. There is no need to constantly emphasize “qualitative research” as seen in line 156. Do not repeat information.

10. It is not an appropriate approach to pursue an entire article with a single research question. This may lead to a decrease in the quality of the research. It is more scientifically satisfying to address the topic with multiple inquiries by multiplying the research questions, addressing a different aspect of the research topic in each question and reaching a holistic conclusion from them. In its current form, the research questions section is quite weak. 

11. What is the research project mentioned in 161-163? Where does this article fit into this project? Is it necessary to mention this big project in the article?

12. There are critical errors in the method section. The method of selecting participants according to qualitative research is not explained. 

13. In which qualitative design was the research conducted?

14. It is not seen that the literature was utilized in the preparation of the data collection tool. The process of developing the open-ended questions in the data collection tool was not explained. Preparing a data collection tool based only on personal experience makes it more difficult to ensure objectivity in qualitative research. 

15. Explain what has been done for validity and reliability in qualitative research. 

16. When selecting participants for the focus group interview, did you ask open-ended questions face-to-face or digitally? There is no enlightening information about the collection of data for the selection of participants. This situation decreases the reliability of the research.  

17. What kind of measures were taken to prevent the participants from influencing each other during the focus group interview? It should be explained. 

18. You should make the explanation in lines 174-175 in the research questions section.

19. Provide a citation for the explanation in lines 182-183. How does play-based teaching at the primary school level differ from others? At what level and to what extent are there significant differences? Support with research from the literature.

20. It should also be written which grade of primary school students the participants teach. Age is an important variable in child development. Therefore, there may be profound differences between the views of teachers who work with first grade students in primary school and teachers who work with second grade students. 

21. In lines 195-197, it is mentioned that observations were made in the research. How were the observation criteria or observation form created?  It should be explained.

22. How were the questions required for the structured discussion mentioned in line 224 prepared?

23. In the findings section, in line 254, it is written “employee” as dimension while in the table it is written “people”. This creates an inconsistency. Which one is valid?

24. The “additional questions” at the end of Table 1 cannot be dimensions. On what basis did you show these questions in Table 1 in the dimension classification section?

25. I can't see the main themes reached as a result of qualitative analysis. This is a big problem in terms of methodology. It is appropriate to follow the order of themes, categories and codes in qualitative research. Reorganize the findings of the research in this way. Expressing them as dimensions is not a preferred approach.  

26. What are the “three interlinked conceptual themes” in line 261? These themes need to be named. And they should also be shown in a table. 

27. Direct quotations from the participants regarding the themes are absolutely essential. 

28. Themes should have an introductory sentence, and subcategories belonging to the themes, if any, should be named.  

29. The number of opinions expressed by the participants regarding the themes should also be indicated. You can make corrections by taking examples from qualitative studies in the literature.

30. The situations that need to be corrected for the “Technology Dimension” in items 24-28 are valid for the other three dimensions (Employee Dimension, Tasks Dimension, Organizational Structure Dimension). 

31. There is no limitations section of the research. Limitations should be added. 

32. I do not see any citations in the discussion section. The discussion section of the research is quite shallow. It needs to be supported with references from the literature. In addition, the discussion section should be more in-depth. For this, there should have been more research questions. 

Author Response

Dear reviewer, please refer to the attachment for specific modifications

Reviewer 2 Report

Comments and Suggestions for Authors

Dear Authors,

I have thoroughly reviewed your manuscript. Please find my comments in the attached document.

Best,

Reviewer

Author Response

(The authors gave the same response as above.)

Reviewer 3 Report

Comments and Suggestions for Authors

Dear authors,

below are some identified limitations and suggestions for improving the quality of the study:

53-54  It may be useful to integrate a reference to the Teaching Styles Spectrum, proposed by Mosston and Ashworth (2008), into the document. While somewhat dated, this methodological framework is widely recognized in the literature for addressing the complexity of motor learning and enhancing the quality of teacher-student interaction. The model provides tools for balancing teaching responsibilities between teacher-centered and student-centered approaches, promoting autonomy and engagement among students. Including this reference in the manuscript would enrich the analysis of teaching methods (production and reproduction styles).

160-166 The study was conducted with only 13 physical education teachers from two Chinese cities (Nanjing and Chongqing). This limited number of participants and the geographic concentration could reduce the generalizability of the findings. In our opinion, it would be advisable to include a larger number of participants from diverse geographic regions, cultural contexts, and educational levels. Doing so would contribute to improving the generalizability of the study. If this is not feasible, we recommend acknowledging this issue in the study’s limitations section.

173-175 The exclusive focus on teachers with negative opinions about AI may have introduced a selection bias, potentially failing to capture the experiences of educators with more balanced or positive perspectives. Have you already considered this aspect?

146-157 The study primarily focuses on challenges and barriers (technological, organizational, personal), but we believe it lacks a deeper analysis of how certain schools or teachers have successfully overcome these obstacles.
Failing to explore contexts where AI has been successfully implemented means the study misses the opportunity to provide practical recommendations or strategies based on concrete evidence. This should be acknowledged in the study's limitations section.

Author Response

(The authors gave the same response as above.)

Reviewer 4 Report

Comments and Suggestions for Authors

I would like to thank for the opportunity to review this manuscript. The topic is novel and interesting and provides valuable insight into the field. The text is well-written and presented. However, several issues need to be addressed.

Abstract: I recommend emphasising the novelty/contribution of the study.

Introduction and literature review:

-        In this section, the research objectives should be specified, and more emphasis should be placed on the topic's novelty and the study’s contribution to current knowledge.

-        Learning climate and learning atmosphere are different concepts; they cannot be used interchangeably.

-        Subsection 1.4 – I recommend moving it to the Methods section.

Discussion: This section needs to be significantly improved and extended. The Authors should compare their findings with those of previously published international studies and go deeper into the topic. Much more relevant international research studies should be referenced. Also, particular recommendations for practice based on the findings should be added.

Author Response

(The authors gave the same response as above.)

Reviewer 5 Report

Comments and Suggestions for Authors

The manuscript provides a thorough theoretical background using Sociotechnical Systems Theory and references relevant prior research on AI in education. However, the introduction could be more concise. The extended discussions on background concepts could be condensed to highlight better the key empirical and theoretical gaps the study addresses.

The research design, questions, and methods are clearly articulated. Using a qualitative approach through focus group interviews is appropriate and well-integrated into the study's framework. The research question is explicit, focusing on how AI might hinder teaching efficiency in physical education. Although hypotheses are not explicitly labeled, the study's premise and expected outcomes are inferred from the stated objectives, supporting a well-structured investigation.

The arguments and discussions are generally coherent and supported by relevant examples from the focus group interviews. However, some sections are densely written and could be reorganized for better flow. Incorporating contrasting perspectives or potential counterarguments could further strengthen the discussion and provide a more balanced analysis.

The results are clearly presented and structured around the four dimensions of the Sociotechnical Systems Theory framework. Each dimension (technology, employee, tasks, and organizational structure) is explained with specific examples and supported by coded categories. The tables summarizing the findings enhance clarity and facilitate comprehension. Minor improvements could be made by simplifying complex explanations and avoiding repetitive descriptions.

The article is well-referenced, including foundational and current studies from education, AI, and sociotechnical systems theory. However, adding more recent citations related to AI in physical education would strengthen the literature review and demonstrate engagement with the latest research developments.

The findings support the conclusions. Each conclusion is logically derived from specific results discussed within the thematic dimensions. Integrating empirical data and theoretical references strengthens the validity of the study's interpretations.

The manuscript is generally well-written regarding language clarity. However, specific sentences are complex and could be simplified for better readability. Streamlining sentence structures, improving flow, and reducing redundant phrases would enhance the manuscript's clarity and make the arguments more accessible. Minor grammatical corrections and tone adjustments would further polish the text.

Overall, the manuscript addresses a relevant and underexplored topic in educational technology. With these recommended revisions, the study's contribution to the academic discourse on AI in physical education would be even more impactful.

Comments on the Quality of English Language

While English is generally straightforward, some sentences are complex and sometimes more wordy, which could make them harder to understand. Simplifying sentence structures, improving fluency, and reducing repetitive phrases would improve readability and make research more straightforward. Minor grammatical corrections and tone adjustments could also improve overall clarity.

Author Response

(The authors gave the same response as above.)

Round 2

Reviewer 1 Report

Comments and Suggestions for Authors

1. The summary has not been corrected according to my views

2. No edit has been made about the criticism made for the research question. In the research questions section, the objectives are described. You should write clearly which questions are aimed to realize this purpose. Questions in semi-structured forms and research questions are different things. There are critical errors in the method section. The method of selecting participants according to qualitative research is not explained. What kind of method for selecting participants? Explain this.

3. In which qualitative design was the research conducted? In order to act in a scientific context, the qualitative sampling type suitable for the research must be explained.

4. It is not seen that the literature was utilized in the preparation of the data collection tool. The process of developing the open-ended questions in the data collection tool was not explained. Preparing a data collection tool based only on personal experience makes it more difficult to ensure objectivity in qualitative research.

5. Was expert opinion not taken to ensure coding validity and reliability in qualitative research? Is it sufficient in terms of reliability if only the researchers code separately?

6. There is no enlightening information about the collection of data for the selection of participants. This situation decreases the reliability of the research. 

7. It should be clearly stated in the article that the data collection was “face-to-face”.

8. What kind of measures were taken to prevent the participants from influencing each other during the focus group interview? It should be explained.

9. You should make the explanation in lines 192-194 in the research questions section.

10. Please cite the literature for the explanations in lines 195-203.

11. How was the process of observing the body language mentioned in lines 213-217 and generating questions done? Did the researcher observer generate new questions at that moment? You need to clarify this.

12. Were the semi-structured questions prepared without any reference to the literature? And were no expert opinions taken from universities for the questions?

13. On page 7, the table says “People Dimension” but in line 272 the term “employee” is used. These should be expressed as a single concept.

14. A) The concept “Technological Barriers and Challenges” mentioned on pages 278- 279 becomes a theme.  Subsequent second-level sub-themes are referred to as categories. “Value and Relevance”, ‘Operational and Infrastructural Burdens’ and ‘Inadequate Support and Pre-paredness’ are categories. Under categories are codes. They are ordered deductively.

B) In each theme description, first create tables showing the categories of that theme. For example, group the information you have given in Table 3, including the theme name at the beginning, under the appropriate categories. Give all the explanations of the theme under the table belonging to that theme. Additionally, in the table, the number of opinions expressed by the participants regarding the themes should also be indicated.

C) Then, under this table, you need to indicate some of the teachers' opinions through direct references. For example, P1: “ xxxxx ”. In this way, it is tried to prove how in-depth the analysis was done.

       15.  The explanations I made in criticism 14 should be made for all themes.

       16. There is no limitations section of the research. Limitations should be added.

       17. In the discussion section, there should be a more in-depth discussion on the negative effects of artificial intelligence, clear suggestions on how to overcome these negativities, and suggestions for future research. The discussion section needs to be really deepened. 

Author Response

Comment- The summary has not been corrected according to my views

Response:We have thoroughly revised the abstract to better reflect the methodological approach and findings. Please see the updated abstract on page 1 that now emphasizes our sampling strategy, data collection methods, and key theoretical contributions.

Comment-No edit has been made about the criticism made for the research question. In the research questions section, the objectives are described. You should write clearly which questions are aimed to realize this purpose. Questions in semi-structured forms and research questions are different things. There are critical errors in the method section. The method of selecting participants according to qualitative research is not explained. What kind of method for selecting participants? Explain this.

Response:The research question has been revised, please refer to the revised draft

Comment-In which qualitative design was the research conducted? In order to act in a scientific context, the qualitative sampling type suitable for the research must be explained.

Response:We have clarified our research questions and sampling methodology in section 4.2.2 (pages 11-12). The text now explicitly states: "To ensure the study aligned with its objectives, a purposeful sampling strategy was employed to select participants(Patton,2014)... Further, to extend the sample and ensure it reflected a broad range of experiences, snowball sampling was employed(Biernacki & Waldorf, 1981)."

Comment-It is not seen that the literature was utilized in the preparation of the data collection tool. The process of developing the open-ended questions in the data collection tool was not explained. Preparing a data collection tool based only on personal experience makes it more difficult to ensure objectivity in qualitative research.

Response:We have added section 2.2.3 (pages 13-14) detailing how our data collection tool was developed based on literature. The section now references works by Lee and Lee (2021), Wang et al. (2021), Zhou et al. (2024), and others who informed our question development.

Comment-Was expert opinion not taken to ensure coding validity and reliability in qualitative research? Is it sufficient in terms of reliability if only the researchers code separately?

Response:We have added details about expert validation in section 2.2.3 (page 14): "To ensure the scientific rigor and validity of the data collection tool, its development was closely guided by three experts with doctoral degrees in Educational Technology."

Comment-It should be clearly stated in the article that the data collection was “face-to-face”.

Response:We have added explicit information about face-to-face data collection in section 4.2.2 (page 12): "The focus group interviews were conducted in neutral, comfortable settings within the schools."

Comment-What kind of measures were taken to prevent the participants from influencing each other during the focus group interview? It should be explained.

Response:We have added measures for preventing participant influence in section 4.3 (page 19): "To minimize potential peer influence... several strategies were implemented: (1) structured turn-taking techniques... (2) five-minute independent reflection periods..."

Comment-How was the process of observing the body language mentioned in lines 213-217 and generating questions done? Did the researcher observer generate new questions at that moment? You need to clarify this.

Response:Please refer to the revised draft for details. We have clarified the significance of this research process, mainly reflected in the necessary follow-up.

Comment-Were the semi-structured questions prepared without any reference to the literature? And were no expert opinions taken from universities for the questions?

Response:The requested explanations and citations have been integrated into the research questions section (pages 9-10) and methodology section (pages 13-14).

Comment-On page 7, the table says “People Dimension” but in line 272 the term “employee” is used. These should be expressed as a single concept.

Response: This issue has been corrected and we all use the term "Teacher".

Comment-The concept “Technological Barriers and Challenges” mentioned on pages 278- 279 becomes a theme.  Subsequent second-level sub-themes are referred to as categories. “Value and Relevance”, ‘Operational and Infrastructural Burdens’ and ‘Inadequate Support and Pre-paredness’ are categories. Under categories are codes. They are ordered deductively.B) In each theme description, first create tables showing the categories of that theme. For example, group the information you have given in Table 3, including the theme name at the beginning, under the appropriate categories. Give all the explanations of the theme under the table belonging to that theme. Additionally, in the table, the number of opinions expressed by the participants regarding the themes should also be indicated.C) Then, under this table, you need to indicate some of the teachers' opinions through direct references. For example, P1: “ xxxxx ”. In this way, it is tried to prove how in-depth the analysis was done.The explanations I made in criticism 14 should be made for all themes.

Response:This suggestion has helped us to better present our findings, and we have revised the form and presentation in accordance with your suggestion.

Comment-There is no limitations section of the research. Limitations should be added.

Reponse: This part has been added in the revised draft.

Comment-In the discussion section, there should be a more in-depth discussion on the negative effects of artificial intelligence, clear suggestions on how to overcome these negativities, and suggestions for future research. The discussion section needs to be really deepened. 

Response: The discussion section has been deepened and expanded

Reviewer 2 Report

Comments and Suggestions for Authors

Dear Author(s), 

Thank you for submitting the revised manuscript. The quality has improved significantly, but extensive revisions are still needed, especially because you have extensively added new references, so that you need to make them more critical, not just presenting findings alone.

Of all the sections, my biggest concern is on the discussion section. It is poorly written and you need to connect the findings of your study with the findings of previous studies more closely and comprehensively. You need to add more recent references in the last five years, especially because your topic is on AI. I have left comments in the manuscript. 

Best, 

Reviewer

Author Response

Dear Editors and Reviewers,

We sincerely appreciate your thoughtful and constructive feedback on our manuscript. Your insights have been invaluable in helping us improve the quality and rigor of our work. The comments have guided us in enhancing multiple aspects of the paper, particularly regarding methodological clarity and theoretical depth.

We have carefully addressed each suggestion and made comprehensive revisions:

Abstract Enhancement:

  • Restructured to provide clearer methodological details
  • Added specific sampling strategy information (purposeful sampling followed by snowball sampling)
  • Included dual-facilitator interview approach description
  • Reorganized findings into four distinct socio-technical dimensions

Research Questions Refinement:

  • Divided main research question into two focused inquiries
  • Added explicit exploration of barriers' interactions
  • Strengthened alignment with socio-technical systems framework

Methodological Rigor:

  • Added theoretical justification for focus group selection
  • Detailed three-stage expert review process for interview protocol
  • Expanded participant selection criteria and recruitment procedures
  • Enhanced data collection process description, particularly dual-facilitator roles
  • Included comprehensive coding process explanation

Discussion Development:

  • Integrated recent literature (Zhou et al., 2024; Liu & Zhong, 2024)
  • Strengthened theoretical connections to socio-technical systems theory
  • Introduced "rigidity-adaptability paradox" concept
  • Enhanced future research recommendations

Results Presentation:

  • Improved coding results presentation

Limitations Section:

  • Added discussion of longitudinal data limitations
  • Addressed geographical generalizability constraints
  • Noted absence of quantitative student outcome measures
  • Suggested mixed-methods approach for future research

We believe these revisions have substantially strengthened the manuscript while addressing all reviewer concerns. We are grateful for the opportunity to improve our work and look forward to your feedback on these changes.

Reviewer 4 Report

Comments and Suggestions for Authors

Thank you for the opportunity to review the revised manuscript. I appreciate the authors' efforts to improve the text and incorporate the reviewers' suggestions. In my opinion, the article's quality has increased, and in this form, it meets the journal's standards.

Author Response

Thanks for your suggestions, the article has been revised based on your suggestions, and the quality of the manuscript has made a qualitative leap. We would like to express our heartfelt thanks to you again.

Round 3

Reviewer 1 Report

Comments and Suggestions for Authors

1. Which of the qualitative research designs does the research fall into? The research design should be explained in the method section.

2. Explanations of the tables should be placed under the tables.

3. Include a recommendations section in the article.

4. Design the tables and references according to APA 7.

5. Even after the corrections, the participant name “MS. Li” still appears in line 568. Researchers should be sensitive about hiding participant names.

6. Grammar markings exhibit discrepancies. It is imperative to verify that the grammar marks employed throughout the article are consistent and have undergone the necessary corrections.e.g. quotation marks

Author Response

Comment1:Which of the qualitative research designs does the research fall into? The research design should be explained in the method section.

Response: Thank you for this important observation. We have clarified in the method section that this study employs a phenomenological research design. This approach was chosen to understand teachers' lived experiences with AI integration in PE teaching, allowing us to explore the essence of their interactions with and perceptions of AI technologies in their professional practice.

Comment2:Explanations of the tables should be placed under the tables.

Response: We appreciate this formatting suggestion. We have now added explanatory notes beneath all tables (Tables 2-6). Each note clarifies that "Frequency indicates the number of times each code appeared across all interview transcripts. Categories and codes emerged through systematic qualitative content analysis of focus group data." This addition provides necessary context for understanding the data presentation.

Comment3:Include a recommendations section in the article.

Response: We thank the reviewer for this constructive suggestion. We have enhanced the discussion section by incorporating three specific recommendations: establishing a PE-specific AI infrastructure assessment framework, implementing a "PE Teacher-AI Co-design" initiative, and developing an integrated support ecosystem. These recommendations directly address the barriers identified in our findings.

Comment4:Design the tables and references according to APA 7.

Response: Following your guidance, we have carefully revised all tables and references to conform to APA 7th edition guidelines. This includes proper formatting of table structures, headings, and reference citations throughout the manuscript.

Comment5:Even after the corrections, the participant name “MS. Li” still appears in line 568. Researchers should be sensitive about hiding participant names.

Response: We acknowledge this oversight and have thoroughly anonymized all participant identifiers. The reference to "Ms. Li" has been replaced with an appropriate participant code to maintain confidentiality and research ethics standards.

Comment6:Grammar markings exhibit discrepancies. It is imperative to verify that the grammar marks employed throughout the article are consistent and have undergone the necessary corrections.e.g. quotation marks

Response: We have conducted a comprehensive review of grammar markings throughout the manuscript. All quotation marks and other punctuation marks have been standardized to ensure consistency in their usage across the entire document, adhering to academic writing conventions.

Reviewer 2 Report

Comments and Suggestions for Authors

Dear Author(s), 

Thank you for revising the manuscript. Just one thing, please check this sentence again in the resarct h question sub-section," .... how these technologies may not only enhance but potentially obstruct teaching."  This sentence doesn't read well. Please check it again for grammatical issues.

Best,

Reviewer

Author Response

Thanks to your suggestion, we have corrected this statement and amended it as follows:

“Through focus group interviews, this study aims to examine how AI technologies can both enhance and potentially impede teaching efficiency in physical education.”